# Synthesis of Novel 2-(Cyclopentylamino)thiazol-4(5*H*)-one Derivatives with Potential Anticancer, Antioxidant, and 11β-HSD Inhibitory Activities

**DOI:** 10.3390/ijms24087252

**Published:** 2023-04-14

**Authors:** Szymon Baumgart, Daria Kupczyk, Aneta Archała, Oliwia Koszła, Przemysław Sołek, Wojciech Płaziński, Anita Płazińska, Renata Studzińska

**Affiliations:** 1Department of Organic Chemistry, Faculty of Pharmacy, Collegium Medicum in Bydgoszcz, Nicolaus Copernicus University in Toruń, 2 Jurasza Str., 85-089 Bydgoszcz, Poland; sz.baumgart@cm.umk.pl; 2Department of Medical Biology and Biochemistry, Faculty of Medicine, Collegium Medicum in Bydgoszcz, Nicolaus Copernicus University in Toruń, 24 Karłowicza Str., 85-092 Bydgoszcz, Poland; dariak@cm.umk.pl; 3Department of Biopharmacy, Medical University of Lublin, 4a Chodźki Str., 20-093 Lublin, Poland; aneta.banach94@o2.pl (A.A.); koszlaoliwia@gmail.com (O.K.); pp.solek@gmail.com (P.S.); wojtek_plazinski@tlen.pl (W.P.); anita.plazinska@umlub.pl (A.P.); 4Jerzy Haber Institute of Catalysis and Surface Chemistry, Polish Academy of Sciences, 8 Niezapominajek Str., 30-239 Krakow, Poland

**Keywords:** 11β-hydroxysteroid dehydrogenase 1, thiazolone derivatives, molecular docking, antitumor activity

## Abstract

In this study, a series of nine new 2-(cyclopentylamino)thiazol-4(5*H*)-one derivatives were synthesized, and their anticancer, antioxidant, and 11β-hydroxysteroid dehydrogenase (11β-HSD) inhibitory activities were tested. Anticancer activity has been assessed using the MTS (MTS: 3-(4,5-dimethylthiazol-2-yl)-5-(3-carboxymethoxyphenyl)-2-(4-sulfophenyl)-2H-tetrazolium) assay against human colon carcinoma (Caco-2), human pancreatic carcinoma (PANC-1), glioma (U-118 MG), human breast carcinoma (MDA-MB-231), and skin melanoma (SK-MEL-30) cancer cell lines. Cell viability reductions, especially in the case of Caco-2, MDA-MB-231, and SK-MEL-30 lines, were observed for most compounds. In addition, the redox status was investigated and oxidative, but nitrosative stress was not noted at a concentration of 500 µM compounds tested. At the same time, a low level of reduced glutathione was observed in all cell lines when treated with compound **3g** (5-(4-bromophenyl)-2-(cyclopentylamino)thiazol-4(5*H*)-one) that most inhibited tumor cell proliferation. However, the most interesting results were obtained in the study of inhibitory activity towards two 11β-HSD isoforms. Many compounds at a concentration of 10 µM showed significant inhibitory activity against 11β-HSD1 (11β-hydroxysteroid dehydrogenase type 1). The compound **3h** (2-(cyclopentylamino)-1-thia-3-azaspiro[4.5]dec-2-en-4-one) showed the strongest 11β-HSD1 inhibitory effect (IC_50_ = 0.07 µM) and was more selective than carbenoxolone. Therefore, it was selected as a candidate for further research.

## 1. Introduction

Pseudothiohydantoin (2-aminoiazol-4(5*H*)-one) is a partially hydrogenated thiazole derivative containing an amino group in the 2-position and a carbonyl group in the 4-position (Figure 1). Pseudothiohydantoin derivatives exhibit a number of biological activities, including antibacterial [1,2,3,4], antifungal [3,5], and antiviral [6,7]. They are also an interesting and promising class of compounds in the search for selective 11β-hydroxysteroid dehydrogenase type 1 (11β-HSD1) inhibitors [8]. 11β-hydroxysteroid dehydrogenase belongs to the short-chain family of nicotinamide adenine dinucleotide (NAD) or nicotinamide adenine dinucleotide phosphate-dependent (NADP-dependent) dehydrogenases/reductases (SDRs). This enzyme exists as 2 isoforms: 11β-hydroxysteroid dehydrogenase type 1 and 11β-hydroxysteroid dehydrogenase type 2 (11β-HSD2). It plays an important role in the peripheral mechanism of cortisol generation. 11β-HSD1 is an NADPH-dependent oxidoreductase that catalyzes the conversion of inactive cortisone to physiologically active cortisol and, to a lesser extent, the reverse reaction. Together with 11β-hydroxysteroid dehydrogenase type 2, it forms a system that regulates the level of cortisol in the body. Inhibiting the activity of 11β-hydroxysteroid dehydrogenase type 1 causes a decrease in cortisol levels, a decrease in adipose tissue mass, insulin resistance, and central obesity, as well as a decrease in total cholesterol. Thus, selective inhibitors have a significant pharmacological potential for type 2 diabetes, obesity, and cardiovascular disease treatment [9].

Pseudothiohydantoin derivatives also exhibit anticancer activity. This research was carried out on many cell lines. Many derivatives have been shown to be cytotoxic to breast cancer cells MCF-7 [3,10,11,12] but also to MDA-MB-231 [5] and BT-474 [3]. Thiazolone derivatives containing the quinolinyl-methylene-group at C-5 also show activity against lung cancer cells H460a, colon cancer (HCT116, SW480, RKO), and osteosarcoma (SJSA1) [13]. The mechanism of pseudothiohydantoin derivatives antitumor activity is reported to be the capacity for various enzyme inhibition inter alia: cyclin-dependent kinase 1 (CDK1) [13], cyclin-dependent kinase 2 (CDK2) inhibition [10], carbonic anhydrase IX (CA IX) [5], and human mitotic kinesin Eg5 [14]. The above reports show that the search for new anticancer drugs among pseudothiohydantoin derivatives may also be an important aspect of research on this group of compounds.

In our earlier studies, we focused on the search for selective inhibitors of 11β-hydroxysteroid dehydrogenase type 1 among 2-aminothiazol-4(5*H*)-one derivatives. We synthesized derivatives substituted in the amino group with allyl [15], methyl [16], isopropyl [17], *tert*-butyl [18], and adamantyl [19] moiety differing in substituents in the 5-position of the thiazole ring. The analysis of the results of in vitro tests combined with molecular modeling allowed us to obtain better and better results, both in inhibiting the activity of this enzyme and in the selectivity of inhibitors. 

Bearing in mind the previously observed relationship between the presence of large hydrophobic groups in the molecule and the inhibition of 11β-HSD1 activity, we decided to synthesize a series of 2-cyclopentylamino derivatives of thiazol-4(5*H*)-one in order to study their activity in this direction. Due to the fact that many compounds containing thiazolone moieties exhibit anticancer activity, we also decided to test the activity of the compounds obtained using selected cell lines.

## 2. Results and Discussion

### 2.1. Chemistry

New 2-(cyclopentylamine)thiazol-4(5*H*)-one derivatives were obtained by reacting 2-cyclopentylthiourea with the appropriate 2-bromo ester (Figure 2). Depending on the ester used, the reactions were carried out using three different procedures. Based on earlier research on pseudothiohydantoin derivatives [19], the synthesis of derivatives with unbranched alkyl substituents at carbon C-5 (**3a**–**3c**) was carried out at room temperature in chloroform (procedure A). 

As a result of the synthesis, compounds **3a**–**3c** were obtained in the hydrobromide form with differentiated yields (5.39–81.63%, Table 1). Procedure A turned out to be ineffective in the case of the synthesis of **3d**–**3e** derivatives; therefore, the synthesis of these compounds was carried out in a sodium methoxide medium at reflux. Under these conditions, the products **3d**–**3e** were obtained with a 9.42–19.88% yield. In order to shorten the reaction time, the synthesis of **3f**–**3i** derivatives containing aromatic substituents and a spiro system of thiazole and alicyclic rings was carried out at a temperature of 150–160 °C in the presence of *N*,*N*-diisopropylethylamine and microwave radiation (procedure C). This was modeled on syntheses previously conducted by Johansson et al. (for detailed description of the procedure, see Section 3.3.3) [20]. Under the presented conditions, it was possible to obtain **3f**–**3g** derivatives with high yields of 71.06–85.48% and **3h**–**3i** compounds with 2.47–27.86% yields in 75 minutes. Lower yields of **3h**–**3i** derivatives are related to the formation of more by-products.

### 2.2. Inhibitory Activity towards 11β-HSD

#### 2.2.1. In Vitro Studies

In this study, all obtained 2-(cyclopentylamino)thiazol-4(5*H*)-one derivatives were tested for their inhibitory effect on two 11β-hydroxysteroid dehydrogenase isoforms: 11β-HSD1 and 11β-HSD2. The results of the in vitro studies are summarized in Table 2. A total of 7 out of 9 new pseudothiohydantoin derivatives (compounds **3c**–**3i**) showed over 50% inhibition of 11β-HSD1 activity at a concentration of 10 µM. 

Among the tested compounds, the strongest inhibitor turned out to be compound **3h** (IC_50_ = 0.07 µM), containing the spiro system of thiazole and cyclohexane rings. This is a better result than that obtained so far for analogous compounds containing other hydrophobic substituents in the amino group. In comparison, the IC_50_ values were 9.35 µM for the isopropylamino derivative [17], 2.5 µM for the allylamino derivative, 1.60 µM [15] for the *tert*-butylamino derivative [18], and 0.31 µM for the adamantylamino derivative [19], respectively. Replacement of the cyclohexyl ring with a cyclobutane ring in compound **3i** resulted in a significant decrease in the inhibitory activity against 11β-HSD1 (IC_50_ = 1.55 µM). Compound **3d** with a spatially large, branched isopropyl substituent and compound **3g** containing a *p*-bromophenyl substituent in position 5 also show high inhibitory activity (IC_50_ = 0.46 µM and 0.18 µM, respectively). Whereas “removal” of the bromine atom from the phenyl group causes a significant decrease in inhibitory activity (compound **3f**, IC_50_ = 1.1 µM). 

The smallest inhibition effect was shown by derivatives containing small aliphatic groups: methyl (**3a**) and ethyl (**3b**). Their percentage of enzyme inhibition was 10.94% and 21.33%, respectively, at the concentration of 10 µM. 

All tested compounds also showed an inhibitory effect on the 11β-HSD2 isoform. They showed inhibition in the range of 36.55% (**3f**) to 46.33% (**3i**) at a concentration of 10 µM. In the case of some previously tested derivatives, lower inhibitory activities were observed in relation to 11β-HSD2 [15,16,17,18,19]. However, due to the relatively weak inhibition of 11β-HSD1 activity by these compounds, these results were not relevant for their use as selective 11B-HSD1 inhibitors. Compounds **3a** and **3b** were the only ones among the compounds obtained to inhibit 11β-HSD2 to a greater extent than 11β-HSD1 activity. Finally, compound **3h** with 90% inhibition of 11β-HSD1 and less than 43% inhibition of 11β-HSD2 turned out to be the most promising. The activity of this compound against 11β-HSD1 is comparable to the known inhibitor, carbenoxolone, but it is characterized by higher selectivity (43% inhibition of 11β-HSD2 at a concentration of 10 µM vs. 55% for carbenoxolone). The obtained results indicate that the compound **3h** is a strong candidate for further research in the search for a drug targeting 11β-HSD1 inhibition.

#### 2.2.2. Molecular Docking

The ligand–protein binding energies found during docking simulations are graphically illustrated in Figure 3A–C. The magnitude of the determined binding energies obtained for the considered set of ligands varies within a relatively narrow range of ~−8.5–−6.6 kcal/mol, indicating, as expected, favorable inhibitor–protein interactions. Five different protein structures were taken into account in the docking procedure in order to account for the inherent conformational flexibility of the protein. The associated scatter of the energies across the considered set of protein structures is relatively small in relation to the magnitude of binding energies and varies from 0.07–0.32 kcal/mol when expressed as standard deviation. This means that the obtained results are free of potential biases resulting from different geometry of the protein backbone., i.e., of a factor that cannot be directly taken into account when using only one protein structure for docking. Independent of that, the flexibility of the sidechains was explicitly accounted for by allowing for their rotation (see the Methods section). When considering the stereoselectivity effects that may influence the binding strength, it was observed that the binding energies corresponding to a given pair of stereoisomers differ very slightly by no more than 0.18 kcal/mol. Moreover, a very high correlation between theoretical binding energies determined for compounds with either R- or S-configuration was obtained (R = 0.996, Figure 3C). Thus, it can be concluded that the stereoconfiguration of the chiral ligand does not significantly affect its binding affinity. (Note that for the 2 least potent compounds, **3a** and **3b**, exhibiting the undetermined but high (>10 µM) IC_50_ value, the fixed IC_50_ = 10 µM equality was assumed).

The order of theoretically determined binding energies correctly reproduces the experimental IC_50_ values for those compounds for which this parameter was measured (Figure 3A,B). This includes the identification of the most (**3i**) and least (**3a**, **3b**) potent compounds. Regarding the complete set of compounds, the correlation of ln(IC_50_) vs. binding energy is apparent, and the associated correlation coefficient varies between 0.874 (compounds with the R-configuration, *p* < 0.002) and 0.896 (compounds with the S-configuration; *p* < 0.001). Moreover, the trend of potencies agrees with the variation of the theoretical binding energies for the complete series of all compounds except **3d** and **3f**. A similar, satisfactory degree of agreement between theoretical energies and ln(IC_50_) is obtained when treating the protein structures independently (correlation coefficients varying between 0.802 and 0.931). 

In view of a satisfactory agreement between the theoretical and experimental results, we carried out a more detailed analysis focused on identifying the structural aspects of ligand-protein interactions. Parallel to binding energies, the results of the docking studies can also be analyzed with respect to the mechanistic interaction patterns that may be significant in the context of the interpretation of the obtained binding energy values and the measured properties. The summary given below relies on analyzing the ligand–protein contacts that occur if the distance between any corresponding atom pair is smaller than the arbitrarily accepted value of 0.4 nm. 

All of the studied ligands bind to the protein structure in nearly the same manner (see Figure 3D), and their orientation in the binding cavity closely resembles that characteristic of other groups of structurally related ligands considered in our previous study [15]. The alternative poses are associated with notably higher energy levels (at least 0.7 kcal/mol above the most favorable energy level) and do not form structurally consistent clusters; thus, they were not included in further analysis. The similarity of the docking poses also includes the stereoisomers of the same compound. A detailed description of the protein–ligand contact pattern is provided below and relies on the most potent compound **3h**. However, as mentioned above, the found interaction pattern is representative of all studied compounds. The graphical illustration of the docking results is given in Figure 3D.

The aliphatic cyclopentyl moiety present in all studied ligands maintains close contact with the sidechains of Tyr183 and Thr124. The attractive interactions maintaining such contacts are the CH–π interactions and hydrophobic contacts involving only the non-polar part of the Thr sidechain, respectively. CH–π interactions are possible also in the case of this fragment of ligand and the Val180 sidechain, which is located slightly further in comparison to Tyr183 and Thr124. The proximity of nicotinamide adenine dinucleotide phosphate (NADP^+^, polar fragments) seems to be an opportunistic result of these interactions. The amine moiety of the ligand molecule located in its central part interacts with the amide group of NADP^+^. Although the distance between potential donors and acceptors is too high to form a stable hydrogen bond (0.38 nm), one cannot exclude that such attractive contact will appear when accounting for conformational heterogeneity driven by the temperature and for the presence of a solvent. The thiazole ring, also located in the central part of the ligand molecule, interacts with Ser170 (hydrogen bonding with its sidechain) and Ala172 (hydrogen bonding with the backbone fragment). The aromatic residue belonging to NADP^+^ is too distant to form stable edge-to-edge π–π stacking, but some smaller energy contributions resulting from this type of interaction can be assumed to exist. Finally, the diverse group of substituents attached to the thiazole ring, of types varying from one ligand to another, exhibit contacts with a number of non-polar amino-acid residues, forming a hydrophobic core and including Leu171, Leu217, Tyr177, and Ala172. Val180, lying at the edge of this cluster, is closer to the opposite site of the ligand molecule but can also support this network of non-polar interactions. The contacts with Tyr177 have a character of the CH–π interaction, whereas those concerning leucines, Ala172, and, possibly, Val180, are hydrophobic interactions, contributing to binding energy by minimizing the area of solvent-exposed non-polar surfaces. 

In summary, the driving force for binding seems to be the hydrophobic interactions of the ligand with non-polar cluster (Leu171, Leu217, and Tyr177) and Tyr183, supported by hydrogen bonding with Ser170 and, possibly, NADP^+^. The divergences between binding energies across the whole group of compounds can be ascribed to the interactions between the substituent attached to the thiazole ring and the region of the above-mentioned non-polar cluster. Given the potential importance of the hydrophobicity of the studied molecules, we also checked how the experimentally determined IC_50_ values correlated with log *p* values. The values of the latter parameter were predicted by using the KOWWIN software (Estimation Programs Interface Suite™ for Microsoft^®^ Windows, v 1.68., United States Environmental Protection Agency, Washington, DC, USA). We have observed an expected negative correlation between either ln(IC_50_) or IC_50_ and log *p*; however, the associated values of *R* were much lower than in the case of correlations with binding energies, i.e., −0.638 and −0.439. This suggests the large contribution of hydrophobicity but also a non-negligible influence of other factors related to more detailed molecular features of both ligand and protein, as discussed above.

Interestingly, when comparing the arrangement of the currently studied series of compounds with our previous results [19], it appears that these structurally related groups of compounds exhibit diverse poses when bound to protein. Namely, compounds considered in ref. [19] interact with non-polar clusters by bulky, aliphatic moiety, common for all ligands, whereas the substituent responsible for varying potency prefers to interact with the vicinity of Tyr183 and Ile121. This divergence can be explained by the large difference in dimensions of the aliphatic substituents in both series. The currently considered compounds contain cyclopentyl substituents of relatively small dimensions, which enables them to accommodate it between Tyr183 and NADP^+^. The alternative substituent [19] is too large to do the same and must be located in the vicinity of Tyr177, where the steric crowd is less intensive. Overall, based on the results of both investigations, it seems that the affinity to the non-polar cluster around Tyr177 is positively correlated with the size of the aliphatic substituent of the ligand molecule.

### 2.3. Anticancer Activity

In the beginning, all the cell line proliferation in response to a wide range of compound concentrations was measured. In detail, we observed statistically significant changes in cell proliferation in a cell type-specific and concentration-dependent manner. We noted cell viability reductions, especially in the case of Caco-2, MDA-MB-231, and SK-MEL-30 lines. The changes were particularly noticeable in higher substances concentration. Substances **3g**, **3h**, and **3i** for Caco-2, **3c**, **3g**, and **3h** for MDA-MB-231, and **3c**, **3f**, and **3g** for SK-MEL-30 were most effective and caused decreases to 20–65% (Figure 4a–i).

The opposite observations were noted for the PANC-1 and U-118 MG cell lines. Treated substances increased cell metabolic activity. The highest increase in proliferation was observed after the administration of **3f**, **3g**, and **3h** for PANC-1 and **3d**, **3f**, and **3h** for U-118 MG, ranging from 115 to 144%. It is worth noticing that only in some experimental sets the highest concentrations of substances applied result in the reduction of PANC-1 and U-118 MG cell viability (Figure 4a–i).

Defective mitochondria function is believed to contribute to mitochondrial DNA (mtDNA) mutation, cell transformation, cancer development, diabetes, and neurodegenerative [21]. The observed decrease in cell metabolic activity may indicate a mitochondrial impairment in terms of adenosine triphosphate (ATP) production and general biogenesis. Such effects may be the beginning of the cellular events leading to apoptosis. In fact, the evidence confirms that there is a direct connection between mitochondrial dynamics and cell life-or-death [22]. An explanation for our results may be redox homeostasis imbalance, a possible modulator of mitochondrial function [23]. Additionally, the important regulation of antioxidant enzyme gene transcription and mitochondrial biogenesis is tightly controlled by reactive oxygen/nitrogen species (ROS/RNS) balance and activation of the nuclear factor-erythroid 2 related factor 2 (Nrf2) [24]. 

### 2.4. Antioxidant Activity

The level of free radicals was assessed for all cell lines tested. A toxic concentration (500 µM) of each compound was selected for assay. The highest increase in the level of reactive oxygen species was noted for the PANC-1 line, except for compound **3a** (Figure 4). Moreover, a high increase in reactive oxygen species (ROS) production was also noted for the U-118 MG (compounds **3a**, **3b**, **3e**, **3h**, and **3i**), Caco-2 (compounds **3b**, **3c**, **3e**, **3g**, and **3i**) and SK-MEL-30 (compounds **3d**–**3h**). Interestingly, there was no increase in ROS for the MDA-MB-231 and BJ lines, except for the compound **3g**, **3h**, and **3i** (Figure 4). In general, ROS are generated by multiple intracellular processes and their overproduction is often observed in response to different toxic agents [25,26]. Such results are reflected in our previous observations and confirm mitochondrial dysfunction. The consequences of oxidative stress or mitochondrial impartment include damage to DNA, proteins, or lipids [27]. However, strategies based on cancer cells elimination by ROS overproduction seem to be effective in anticancer therapy, and in this context, our results potentially set directions for further research in the future.

In the case of reactive nitrogen species, most of the cells remained unaffected; however, the highest increase was observed for the Caco-2 line (compounds **3a**–**3c**). For the other lines, an increase in RNS was noted only for BJ (compound **3b**), U-118 MG (compounds **3a**, **3d**, and **3h**), PANC-1 (compound **3e**), and MDA-MB-231 cell line (compound **3a**) (Figure 5). Nitric oxide molecules act as a messenger with diverse functions and play an essential role in several physiological and pathological processes [28]. However, chronic expression of NO contributes to direct tissue toxicity and is associated with various carcinomas and inflammatory conditions [29].

In addition, the level of reduced glutathione was depleted in most experimental sets, except for BJ (compound **3a**) and U-118 MG (compounds **3a**, **3b**, **3c**, and **3e**). An especially strong reduction was observed for compound **3g** in all cell lines tested (Figure 5). As expected, the antioxidant defense system was activated and is always linked to the redox state of the cell lines tested. This system regulates ROS levels and protects cells from oxidative damage [30]. Importantly, upon a high increase in ROS level, antioxidant defenses may promote cell death. Taken together, redox reactions are the core of mitochondrial function, so cancer therapy targeting these organelles may prove to be an attractive approach.

## 3. Materials and Methods

### 3.1. General Information

Microwave reactions were carried out using the MAGNUM V2 microwave reactor/mineralizer from ERTEC (Poland). ^1^H- (Appendix A) and ^13^C-NMR (Appendix A)—spectra the Bruker Avance 400 and 700 apparatus (TMS as an internal standard, Bruker Billerica, MA, USA). HRMS (high-resolution mass spectrometry) (Appendix A)—Synapt G2 Si mass spectrometer (Waters). The measurement results were processed with MassLynx 4.1 software (Waters, Warsaw, Poland).

### 3.2. Reagents and Solvents

*Solvents:* chloroform, diethyl ether, dimethylsulfoxide, ethyl alcohol, and methyl alcohol (Avantor Performance Materials Poland S.A., Gliwice, Poland). 

*Reagents for synthesis: N*-cyclopenthylthiourea 95% (Enamine, Ukraine), 2-bromo esters: ethyl 2-bromopropionate 99%, 2-bromobutyrate 98%, 2-bromovalerate 99%, 2-bromo-3-methylbutyrate 95%, 2-bromoisobutyrate 98%, 2-bromophenyl acetate 97%, 2-bromo(4-bromophenyl) acetate 97%, bromocyclobutane carboxylate 95%, and methyl 1-bromocyclohexane carboxylate 97% (Alfa Aesar, Kandel, Germany, Acros Organic, Geel Belgium, Sigma-Aldrich, Poznań, Poland).

*Auxiliary reagents: N*-ethyldiisopropylamine 99% (Alfa Aesar, Kandel, Germany), hydrochloric acid, magnesium sulphate, sodium, and sodium hydroxide (Avantor Performance Materials Poland S.A., Gliwice, Poland).

*Thin layer chromatography (TLC) and column chromatography:* 5 × 10 cm silica gel TLC plates coated with F-254 (Merck, Darmstadt, Germany). 

*Column chromatography:* silica gel MN kieselgel 60 M with 0.04–0.063 mm grain diameter (Macherey-Nagel, Oensingen, Switzerland).

*11β-HSD1 assays:* carbenoxolone (disodium salt) (Sigma-Aldrich, Poznań, Poland), cortisone, NADPH tetrasodium salt, phosphate buffer powder (Sigma-Aldrich, Poznań, Poland), pooled human liver microsomes, mixed gender, 1 mL, 20 mg/mL Lot No. 1410013—XenoTech, Cortisol Elisa Ref DkO001 Lot No. 5671A (DiaMetra, Spello, Italy), ELISA Kit for 11-Beta-Hydroxysteroid Dehydrogenase Type 1 Lot No. L211008799 (Cloud-Clone Corp., Wuhan, China), and PBS Lot No. H161008 (Pan Biotech, Aidenbach, Germany).

*11β-HSD2 assays:* 18-beta-glycyrrhetinic acid (Acros Organic, Geel, Belgium), cortisone, NAD cofactor, phosphate buffer powder (Sigma-Aldrich, Poznań, Poland), Human Kidney Microsomes, mixed gender, 0.5 mL, 10 mg/mL Lot No. 1710160 XenoTech, Cortisol Elisa Ref DkO001 Lot No. 5671A (DiaMetra, Spello, Italy), Enzyme-Linked Immunosorbent Assay (ELISA) Kit for 11-Beta-Hydroxysteroid Dehydrogenase Type 2 Lot No. L191113457 (Cloud-Clone Corp., Wuhan, China), and PBS Lot No. H161008 (Pan Biotech, Aidenbach, Germany).

### 3.3. General Procedures of Synthesis

The course of the reaction and the purity of the obtained compounds were controlled using TLC chromatography with ethyl acetate (**3a**-**3c**) and a mixture of chloroform/ethanol 9:1 (**3d**-**3i**) as eluents.

#### 3.3.1. Procedure A (Synthesis of Compounds **3a**-**3c**) 

Firstly, 0.002 mol of *N*-cyclopentylthiourea (**1**) and 0.0022 mol of the corresponding 2-bromo ester (**2a**-**2c**) were added to 27 mL of chloroform. The reaction mixture was stirred at room temperature for 14–21 days (TLC control). The solvent was evaporated under reduced pressure. The crude product was crystallized from ethanol (**3a**) or diethyl ether (**3b**, **3c**). The crystallized precipitate of the hydrobromide products was dissolved in water and neutralized with NaOH to pH 7-8. The product was extracted in chloroform, then concentrated and crystallized in ethyl ether.

2-(cyclopentylamino)-5-methylthiazol-4(5*H*)-one (**3a**)—Yield: 5.39%. M.p. 208.1–213 °C. ^1^H-NMR (700 MHz, CDCl_3_, δ ppm, J Hz): 12.32 (s, 1H, N^+^H), 4.33 (q, 1H, C^5^-H, 7.7), 3.89 (s, 1H, NH), 2.04–2.16 (m, 2H, C_5_H_9_), 1.90–2.00 (m, 4H, C_5_H_9_), 1.71–1.78 (m, 2H, C_5_H_9_), 1.81 (d, 1H, CH_3_, 7.7). ^13^C-NMR (100 Hz, CDCl_3_, δ ppm): 171.85 (C-4), 171.65 (C-2), 59.73 (1C, C_5_H_9_), 44.83 (C-5), 32.53 (1C, C_5_H_9_) 32.48 (1C, C_5_H_9_), 23.71 (2C, C_5_H_9_), 18.06 (CH_3_). HR-MS *m*/*z* 199.0905 [M^+^ + 1] (calcd for C_9_H_14_N_2_OS: 199.0905). 

2-(cyclopentylamino)-5-ethylthiazol-4(5*H*)-one (**3b**)—Yield: 69.29%. M.p. 211.5–213.7 °C. ^1^H-NMR (700 MHz, CDCl_3_, δ ppm, J Hz): 12.23 (s, 1H, N^+^H), 4.33 (dd, 1H, C^5^-H, 4.2, 7.7), 3.93 (s, 1H, NH), 2.23–2.31 (m, 1H, C^5^-CH_B_CH_3_), 2.06–2.15 (m, 3H, C_5_H_9_ (2H), C^5^-CH_A_CH_3_), 1.91–2.01 (m, 4H, C_5_H_9_), 1.71–1.80 (m, 3H, C_5_H_9_), 1.14 (t, 3H, CH_3_). ^13^C-NMR (100 Hz, CDCl_3_, δ ppm): 171.57 (C-4), 171.29 (C-2), 58.01 (1C, C_5_H_9_), 50.67 (C-5), 34.14 (1C, C_5_H_9_), 31.37 (1C, C_5_H_9_), 24.50 (1C, C_5_H_9_), 23.81 (1C, C_5_H_9_), 20.37 (CH_2_), 13.29 (CH_3_). HR-MS *m/z* 213.1062 [M^+^ + 1] (calcd for C_10_H_17_N_2_OS: 213.1062). 

2-(cyclopentylamino)-5-propylthiazol-4(5*H*)-one (**3c**)—Yield: 81.63%. M.p. 192-194 °C. ^1^H-NMR (700 MHz, CDCl_3_, δ ppm, J Hz): 11.86 (s, 1H, N^+^H), 11.68 (d, 1H, N^+^H, 7.7), 4.89 (s, 1H, N-H), 4.42 (dd, 1H, C^5^-H, 4.9, 9.1), 4.23 (dd, 1H, C^5^-H, 4.9, 9.1), 3.94 (s, 1H, N-H), 2.06–2.26 (m, 5H, C^5^-CH_B_CH_2_CH_3_, C_5_H_9_), 1.95–2.01 (m, 1H, C^5^-CH_A_CH_2_CH_3_), 1.82-1.94 (m, 4H, C_5_H_9_), 1.64–1.77 (m, 2H, C_5_H_9_), 1.42–1.61 (m, 2H, C^5^-CH_2_CH_2_CH_3_), 1.03 (t, 3H, CH_3_, 7), 0.97 (t, 3H, CH_3_, 7). ^13^C-NMR (100 Hz, CDCl_3_, δ ppm): 172.25 (C-4), 171.42 (C-2), 59.73 (1C, C_5_H_9_), 51.01 (C-5), 34.07 (1C, C_5_H_9_), 32.53 (1C, C_5_H_9_), 24.50 (1C, C_5_H_9_), 23.97 (1C, C_5_H_9_), 23,73 CH_2_), 20.30 (CH_2_), 13.29 (CH_3_). HR-MS *m/z* 227.1221 [M^+^ + 1] (calcd for C_11_H_19_N_2_OS: 227.1218). 

#### 3.3.2. Procedure B (Synthesis of Compounds **3d**-**3e**)

Firstly, 0.002 mol of *N*-cyclopentylthiourea (**1**) and 0.0022 mol of the corresponding 2-bromo ester (**2d**-**2e**) were added to the solution of sodium methoxide (prepared with 0.092 g of sodium in 8 mL of anhydrous methanol). The mixture was refluxed for 7 (**3d**) and 14 days (**3e**), respectively. After evaporation of the solvent, the resulting oily post-reaction mixture was dissolved in 20 mL of water and neutralized with HCl to pH 7–8. The reaction mixture was extracted with chloroform (4 × 20 mL), then dried over magnesium sulfate, and the solvent evaporated. The concentrated mixture was purified by column chromatography (eluent chloroform/ethanol 9:1). The pure main products were crystallized from diethyl ether.

2-(cyclopentylamino)-5-isopropylthiazol-4(5*H*)-one (**3d**)—Yield: 81.63%. M.p. 135-138 °C. ^1^H-NMR (700 MHz, CDCl_3_, δ ppm, J Hz): 4.21 (d, 1H, C^5^-H, 3.5), 3.82–3.90 (m, 1H, N-H), 2.55–2.63 (m, ^1^H, C^5^-CH), 2.00–2.13 (m, 2H, C_5_H_9_), 1.83–1.98 (m, 4H, C_5_H_9_), 1.83–1.66 (m, 3H, C_5_H_9_), 1.08 (d, 3H, CH_3B_, 7), 0.92 (d, 3H, CH_3B_, 7). ^13^C-NMR (100 Hz, CDCl_3_, δ ppm): 186.43 (C-4), 181.02 (C-2), 62.82 (1C, C_5_H_9_), 57.86 (C-5), 32.63 (2C, C_5_H_9_), 30.54 (1C, CH(CH_3_)_2_), 23.99 (2C, C_5_H_9_), 22.06 (CH(CH_3A_)_2_), 16.34 (CH(CH_3B_)_2_). HR-MS *m/z* 227.1222 [M^+^ + 1] (calcd for C_11_H_19_N_2_OS: 227.1218).

2-(cyclopentylamino)-5,5-dimethylthiazol-4(5*H*)-one (**3e**)—Yield: 9.42%. M.p. 178–181 °C. ^1^H-NMR (700 MHz, CDCl_3_, δ ppm, J Hz): 3.69–3.78 (m, 1H, N-H), 1.99–2.12 (m, 2H, C_5_H_9_), 1.81–1.97 (m, 4H, C_5_H_9_), 1.52–1.70 (m, 4H, C_5_H_9_), 1.64 (s, 6H, 2xCH_3_). ^13^C-NMR (100 Hz, CDCl_3_, δ ppm): 190.63 (C-4), 179.38 (C-2), 59.77 (1C, C_5_H_9_), 57.86 (C-5), 32.47 (2C, C_5_H_9_), 27.86 (2C, CH(CH_3_)_2_), 23.90 (2C, C_5_H_9_). HR-MS *m/z* 213.1061 [M^+^ + 1] (calcd for C_10_H_17_N_2_OS: 213.1062). 

#### 3.3.3. Procedure C (Synthesis of Compounds **3f**-**3i**) 

Firstly, 0.02 mol of *N*-cyclopentylthiourea (**1**), 0.0022 mol of the corresponding 2-bromo ester (**2f**–**2g**), 0.374 mL of *N*,*N*-diisopropylethylamine, and 2 mL of anhydrous ethanol were added to the reaction vessel. The mixture was heated in a microwave reactor in two cycles (I—15 min, 150–155 °C, II—1 h, 155–160 °C. After heating, the solvent was evaporated, the residue was dissolved in 20 mL of water, and the pH was measured. The neutral fraction was extracted with chloroform (4 × 20 mL). The organic fraction was dried over MgSO_4_, filtered, and evaporated. The main product was isolated from the reaction mixture by column chromatography (chloroform/ethanol 9:1).

2-(cyclopentylamino)-5-phenylthiazol-4(5*H*)-one (**3f**)—Yield: 71.06%. M.p. 183–184 °C. ^1^H-NMR (700 MHz, CDCl_3_, δ ppm, J Hz): 7.32–7.44 (m, 5H, C_6_H_5_), 4.40 (s, 1H, C^5^-H), 3.80–3.90 (m, 1H, N-H), 2.03–2.13 (m, 2H, C_5_H_9_), 1.78–2.02 (m, 4H, C_5_H_9_), 1.53–1.77 (m, 3H, C_5_H_9_). ^13^C-NMR (100 Hz, CDCl_3_, δ ppm): 181.75 (C-4), 178.63 (C-2), 134.79 (1C_Ph_), 129.24 (1C_Ph_), 128.82 (1C_Ph_), 128.41 (2C_Ph_), 58.34 (1C, C_5_H_9_), 57.24 (C-5), 32.65 (2C, C_5_H_9_), 23.91 (2C, C_5_H_9_). HR-MS *m/z* 261.1065 [M^+^ + 1] (calcd for C_14_H_17_N_2_OS: 261.1062). 

2-(cyclopentylamino)-5-(4-bromophenyl)thiazol-4(5*H*)-one (**3g**)—Yield: 85.48%. M.p. 220 °C (dec.). ^1^H-NMR (700 MHz, DMSO, δ ppm, J Hz): 7.47–7.54 (m, 2H, C_6_H_4_), 7.22–7.27 (m, 2H, C_6_H_4_), 5.14 (s, 1H, C^5^-H), 3.79–3.87 (m, 1H, N-H), 2.01–2.16 (m, 4H, C_5_H_9_), 1.80–1.99 (m, 4H, C_5_H_9_), 1.51–1.77 (m. 3H, C_5_H_9_). ^13^C-NMR (100 Hz, CDCl_3_, δ ppm): 185.43 (C-4), 180.97 (C-2), 135.01 (1C, C_6_H_4_), 132.15 (2C, C_6_H_4_), 129.98 (2C, C_6_H_4_), 122.51 (1C, C_6_H_4_), 57.91 (1C, C_5_H_9_), 57.74 (C-5), 32.65 (2C, C_5_H_9_), 23.96 (2C, C_5_H_9_). HR-MS *m/z* 339.0170 [M^+^ + 1] (calcd for C_14_H_16_N_2_OS^79^Br: 339.0167). 

2-(cyclopentylamino)-1-thia-3-azaspiro[4.5]dec-2-en-4-one (**3h**)—Yield: 2.47%. M.p. 207.7–209.1 °C. ^1^H-NMR (700 MHz, CDCl_3_, δ ppm, J Hz): 3.75 (m, 1H, N-H), 1.98–2.13 (m, 4H, C_5_H_9_, C_5_H_10_), 1.86–1.97 (m, 6H, C_5_H_9_, C_5_H_10_), 1.82–1.85 (m, 1H, C_5_H_10_), 1.69–1.82 (m, 2H, C_5_H_10_), 1.51–1.65 (m, 3H, C_5_H_9_), 1.32–1.46 (m, 3H, C_5_H_10_). ^13^C-NMR (100 Hz, CDCl_3_, δ ppm): 190.73 (C-4), 180.65 (C-2), 68.96 (C-5), 58.00 (1C, C_5_H_9_), 36.63 (2C, C_5_H_10_), 32.51 (2C, C_5_H_10_), 25.14 (2C, C_5_H_10_), 24.94 (1C, C_5_H_10_), 23.95 (2C, C_5_H_9_). HR-MS *m/z* 253.1377 [M^+^ + 1] (calcd for C_13_H_21_N_2_OS: 253.1375). 

6-(cyclopentylamino)-5-thia-7-azaspiro[3.4]oct-6-en-8-one (**3i**)—Yield: 27.86%. M.p. 202.4–203.1 °C. ^1^H-NMR (700 MHz, CDCl_3_, δ ppm, J Hz): 3.68–3.77 (m, 1H, N-H), 2.75–2.87 (m, 2H, C_3_H_6_), 2.46–2.56 (m, 2H, C_3_H_6_), 2.25–2.34 (m, 1H, C_3_H_6_), 1.98–2.12 (m, 3H, C_5_H_9_, C_3_H_6_), 1.84–1.97 (m, 4H, C_5_H_9_), 1.49–1.77 (m, 3H, C_5_H_9_). ^13^C-NMR (100 Hz, CDCl_3_, δ ppm): 190.76 (C-4), 179.74 (C-2), 60.70 (C-5), 57.86 (1C, C_5_H_9_), 34.19 (2C, C_3_H_6_), 32.53 (2C, C_5_H_9_), 23.94 (2C, C_5_H_9_) 16.89 (1C, C_3_H_6_). HR-MS *m/z* 225.1062 [M^+^ + 1] (calcd for C_11_H_17_N_2_OS: 225.1062). 

### 3.4. Inhibition of 11β-HSD Assays 

#### 3.4.1. 11β-HSD1 

Human liver microsomes were used as a source of 11β-HSD1 enzyme to study the inhibitory effect of **3a**–**3i** on the conversion of cortisone to cortisol [31]. Standard 96-well microplates were filled with reagent mixture: cortisone/NADPH (20 μL, to achieve the final concentration of 200 nM/2 μM), microsomes (10 μL, 1.13 μg of 11β-HSD1 in 1 mL) solution in PBS (final amount of 2.5 μg), phosphate buffer (60 μL, pH 7.4), and 10 µL of compounds 3a-3j dissolved in the mixture containing 1% of DMSO and 99% of water. The resulting solution with a final volume of 100 µL was incubated for 2.5 h at 37 °C. To stop the reaction, 10 µL of a 100 µM solution of 18β-glycyrrhetinic acid in PBS was added. The level of cortisol obtained in the reaction was measured by commercially available ELISA kit.

#### 3.4.2. 11β-HSD2

Human liver microsomes were used as a source of 11β-HSD2 enzyme to study the inhibitory effect of **3a**–**3i** on the conversion of cortisol to cortisone. Standard 96-well microplates were filled with reagent mixture: cortisol/NAD+ (20 μL, to achieve the final concentration of 200 nM/2 μM), microsomes (10 μL, 0.127 μg of 11β-HSD2 in 1 mL) solution in PBS (final amount of 2.5 μg), phosphate buffer (60 μL, pH 7.4), and 10 µL of compounds 3a-3j dissolved in the mixture containing 1% of DMSO and 99% of water. The resulting solution with a final volume of 100 µL was incubated for 2.5 h at 37 °C. To stop the reaction, 10 μL of a 100 μM solution of carbenoxolone in PBS was added. The level of unreacted cortisol was measured by a commercially available ELISA kit.

#### 3.4.3. Determination of IC_50_

Calibration curves to determine IC_50_ values for compounds **3a**–**3i** were obtained using their solutions at concentrations of 0.078, 0.156, 0.3125, 0.625, 1.25, 2.5, 5.0, and 10.0 µM and using standard procedure and conditions as described in the previous sections. As a control, analogous tests without the addition of inhibitors were used. Half the inhibitory concentration (causing 50% reduction of cortisol or cortisone) was read from the chart.

### 3.5. Molecular Docking 

Ligand molecules (see Table 1) were drawn manually by using the Avogadro 1.1.1 software [32] and optimized within the UFF force field [33] (5000 steps, conjugate gradient algorithm). Six out of nine ligands are chiral compounds; in their cases, docking was performed separately for each stereoisomer. The flexible, optimized ligand molecules were docked into the binding pocket of the five protein structures found in the PDB database (www.rcsb.org; accessed on 13 April 2023): PBD: 3crz, 3qqp, 4bb5, 4c7j, and 4hfr. The crystal resolutions of the particular structures are equal to 0.235, 0.272, 0.220, 0.216, and 0.273 nm, respectively. The AutoDock Vina software [34] was applied for docking simulations. The procedure of docking was carried out within the cuboid region of dimensions of 22 × 22 × 22 Å^3^, which covers the originally co-crystallized ligands present in the PDB structures as well as the closest amino acid residues that exhibit contact with those ligands (including the NADP^+^ molecule, present in all PDB entries and maintaining contact with co-crystalized inhibitors). All the default procedures and algorithms implemented in AutoDock Vina were applied during docking procedure. In addition to the flexibility of the ligand molecules, the rotation of selected sidechains (Leu126, Leu171, Tyr177, Tyr183, Leu215, Leu217, Thr222, Thr124, and Val180) in the proximity of the co-crystalized ligands was allowed as well. The predicted binding energies were averaged over five protein structures, and each stereoconfiguration of the ligand was treated separately (results for non-chiral ligands were included in the analysis in combination with both stereoconfigurations). The more favorable binding mode is correlated with the lower binding energy value; only the lowest energy values corresponding to the given ligand were accounted for during subsequent analysis. The visual inspection of the location and orientation of the docked ligands in order to control the uniformity of the binding pattern was performed. 

The docking methodology was initially validated by docking simulations of several ligand molecules originally included in the protein structures (i.e., PDB:4p38, 3czr, 3qqp, and 4hfr). The description of the validation procedure and the graphical illustration of its results can be found in Ref. [15]. Here, we only mention that the accepted methodology proved accurate enough to recover the original position of the bound ligand.

### 3.6. Cell Culture

The BJ (CRL-2522), Caco-2 (HTB-37), PANC-1 (CRL-1469), U-118 MG (HTB-15), SK-MEL-30 (ACC 151), and MDA-MB-231 (HTB-26) cell lines were purchased from ATCC (Manassas, VA, USA) or DSMZ (Braunschweig, GERMANY). Cells were cultured in accordance with manufacturers' recommendations in a DMEM, MEM, or RPMI-1640 (Sigma, Saint Louis, MO, USA; Gibco, Waltham, MA, USA), supplemented with 10% FBS, except for Caco-2 (20%) (Thermo Fisher Scientific, Waltham, MA, USA) and 1% antibiotic mix solution (penicillin, streptomycin; Thermo Fisher Scientific, Waltham, MA, USA) in a humidified atmosphere with 5% CO_2_ at 37 °C. All cell lines were cultured in T75 cell culture flasks or cell culture dishes to reach 80–90% confluency and seeded at a standard density of 5 × 10^3^ cells/0.32 cm^2^, except for BJ cells at 2 × 10^3^ cells/0.32 cm^2^.

### 3.7. Metabolic Activity

The MTS ((MTS: 3-(4,5-dimethylthiazol-2-yl)-5-(3-carboxymethoxyphenyl)-2-(4-sulfophenyl)-2H-tetrazolium)) assay was performed following previously reported method [35]. Briefly, cells were seeded at a standard density into 96-well flat-bottom plates and incubated for 24 h, followed by the treatment. The various concentrations of compounds were prepared and treated for the next 72 h. MTS solution was added for **3h** and incubated at standard conditions, protected from light. Finally, the absorbance was read at 590 nm and 620 nm wavelengths using a Synergy H1 microplate reader (BioTek, Santa Clara, CA, USA). The results are presented as %, while untreated control cells were considered 100%.

### 3.8. Antioxidant Activity

The assay was performed following previously reported method [36]. The cells (BJ, Caco-2, PANC-1, U-118 MG, SK-MEL-30, and MDA-MB-231) were seeded at an established density into 96-well black plates. Measurement of intracellular superoxide, nitric oxide, and free thiol levels were assessed with the use of fluorogenic probes: dihydroethidium (Cayman Chemical, Ann Arbor, MI, USA, #12013), DAF-2 diacetate (Cayman Chemical, Ann Arbor, MI, USA, #85165), and Thiol Tracker Violet (Thermo Scientific, Waltham, MA, USA, #T10095). After compound treatment (72 h), the cells were probed at a final concentration of 5 µM each for 15 min at 37 °C. Quantifications were taken with a BioTek–Synergy H1 microplate reader, and the results are presented as mean values + SD.

### 3.9. Statistical Analysis

Statistical analysis of the experimental data was carried out using GraphPad Prism ver. 6.0. All results are presented as means ± standard deviation. Differences between control and test samples were assessed with one-way ANOVA of variance with Dunnett post hoc test. A *p*-value of <0.05 was considered statistically significant between groups and is displayed as: * *p* < 0.05; ** *p* < 0.01; and ****p* < 0.001.

## 4. Conclusions

In this study, nine new 2-(cyclopentylamino)thiazol-4(5*H*)-one derivatives, which could act as inhibitors of 11β-hydroxysteroid dehydrogenase isoenzymes, were synthesized and evaluated. All obtained compounds showed different inhibitory activity against both isoforms of 11β-HSD (10.94–90.49% for 11β-HSD1 and 36.55–46.33% for 11β-HSD2). Compound **3h** containing a spiro system of cyclohexyl and thiazole rings turned out to be a stronger and more selective 11β-HSD1 inhibitor (IC_50_ = 0.07 µM) than the most known 11β-HSD1 inhibitor—carbenoxolone. This result makes compound **3h** a strong candidate for further research. The results of molecular docking of the obtained compounds with 11β-HSD1 are consistent with the results of in vitro tests.

The second objective of this study was to evaluate the anti-proliferative activity of the obtained compounds. All thiazolone derivatives showed antiproliferative activity, especially against three tumor cell lines: Caco-2, MDA-MB-231, and SK-MEL-30. Among the obtained compounds, the **3g** derivative showed the greatest anticancer activity, as it caused the greatest decrease in viability among all five cancer cell lines. 

## Figures and Tables

**Figure 1 ijms-24-07252-f001:**
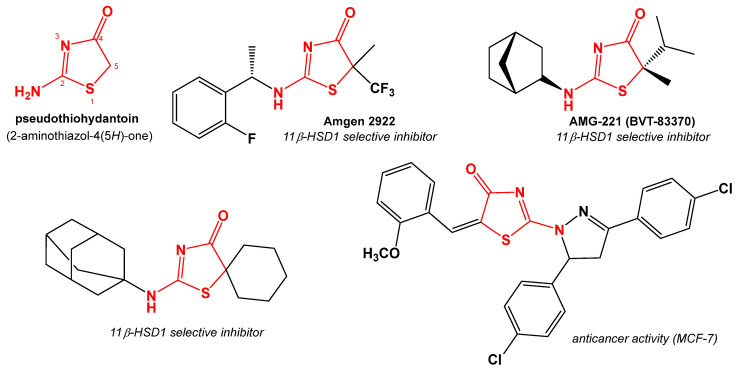
Structures of selected biologically active pseudothiohydantoin derivatives.

**Figure 2 ijms-24-07252-f002:**
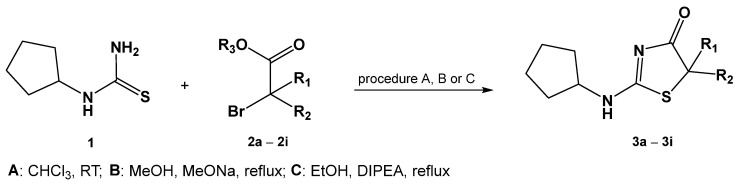
Synthesis of 2-(cyclopentylamino)thiazol-4(5*H*)-one.

**Figure 3 ijms-24-07252-f003:**
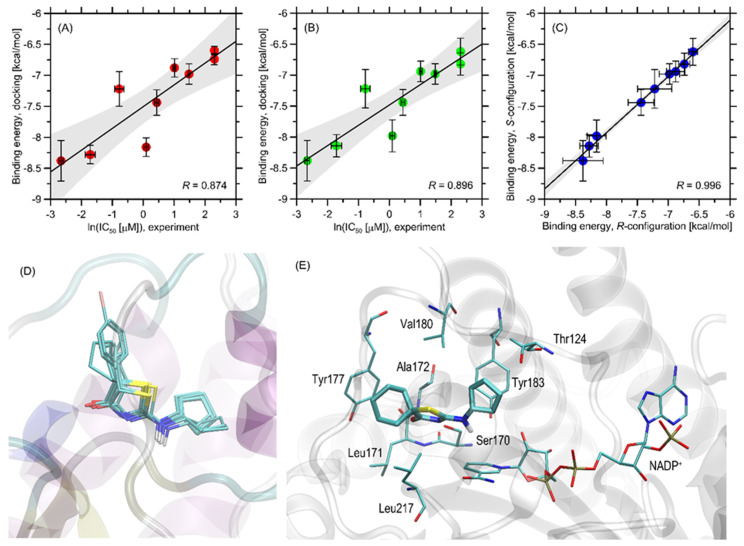
(**A**) The correlation between the binding energies calculated for compounds with R-configuration at the chiral center, averaged over the five protein structures used for docking and the corresponding (experimental) values of IC_50_ (recalculated as log(IC_50_)). The horizontal bars represent the standard deviation values within the set of binding energies. The solid line represents the linear regression (with 95% confidence intervals filled in grey) (**B**) The same as in (**A**) but for compounds with the S-configuration. (**C**) The correlation of theoretical binding energies determined for compounds with the R- and S-configurations. (**D**) The superposed positions of all studied ligands of S-configuration (stick representation) in the binding cavity of the PDB:4bb5 structure. All these positions were identified as the optimal ones during the docking procedure. (**E**) The energetically favorable location of the **3h** compound molecule bound to the PDB:4bb5 structure. The ligand molecule is shown as thick sticks, whereas all the closest amino-acid residues (<0.4 nm) are represented by thin sticks. The NADP^+^ molecule also present in the protein crystal structure is included. The description of the interaction types is given in the text.

**Figure 4 ijms-24-07252-f004:**
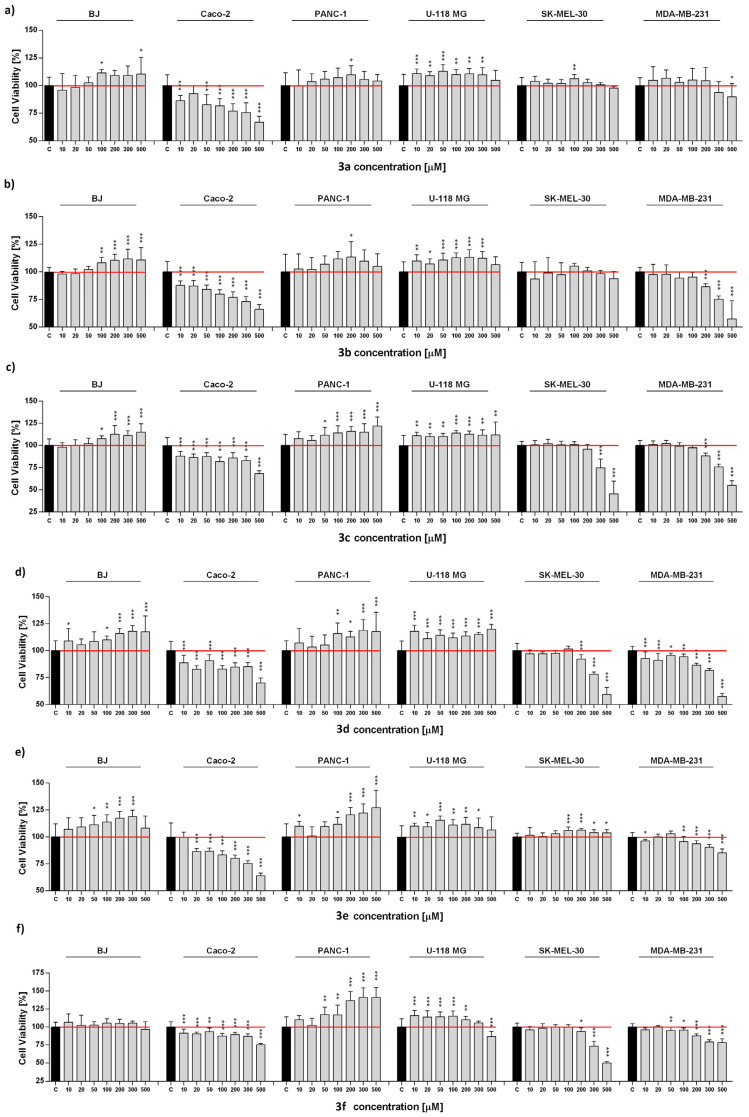
(**a**–**i**) Cell metabolic activity changes in normal and cancer cell lines. Cells were treated with different compounds (**3a**–**3i**) in a wide range of concentrations for 72 h, and viability was assessed by MTS assay. Bars indicate mean value and SD, n = 8, *** *p* < 0.001, ** *p* < 0.01, * *p* < 0.05 (one-way ANOVA and Dunnett’s a posteriori test). For the numerical results, see S28 point in Appendix A section.

**Figure 5 ijms-24-07252-f005:**
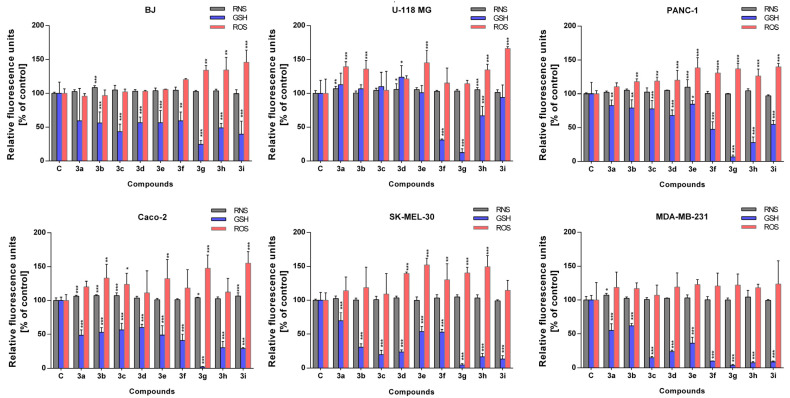
Redox homeostasis disruption in normal and cancer cell lines. Cells were treated with different compounds (**3a**–**3i**) in a wide range of concentrations for 72 h and redox balance was assessed by superoxide (ROS), nitric oxide (RNS) and thiol (GHS) fluorogenic probes. Bars indicate mean value and SD, n = 8, *** *p* < 0.001, ** *p* < 0.01, * *p* < 0.05 (one-way ANOVA and Dunnett’s a posteriori test).

**Table 1 ijms-24-07252-t001:** The yields and melting points of 2-(cyclopentylamino)thiazol-4(5*H*)-one.

No.	R_1_	R_2_	Procedure	Isolated Yield [%]	M.p. (°C)
**3a**	H	CH_3_	A	5.39 *	127.1–128.9
**3b**	H	C_2_H_5_	A	69.29 *	132.1–134.0
**3c**	H	C_3_H_7_	A	81.63 *	112.4–114.0
**3d**	H	CH(CH_3_)_2_	B	19.88	135.0–138.0
**3e**	CH_3_	CH_3_	B	9.42	179.0–181.0
**3f**	H	C_6_H_5_	C	71.06	183.0–184.0
**3g**	H	C_6_H_4_*p*-Br	C	85.48	220 (dec.)
**3h**	-(CH_2_)_5_-		C	2.47	207.7–209.1
**3i**	-(CH_2_)_3_-		C	27.86	202.4–203.1

* for the hydrobromide.

**Table 2 ijms-24-07252-t002:** Inhibitory activity of 2-(cyclopentylamino)thiazol-4(5*H*)-one derivatives.

No.	R_1_	R_2_	% of 11β-HSD1 Inhibition 10 μM	IC_50_ 11β-HSD1 [µM]	% of 11β-HSD2 Inhibition 10 μM
**3a**	H	CH_3_	10.94 ± 4.08	>10	40.27 ± 0.65
**3b**	H	C_2_H_5_	21.33 ± 3.29	>10	39.70 ± 0.07
**3c**	H	C_3_H_7_	56.58 ± 6.11	2.75 ± 0.12	37.70 ± 6.07
**3d**	H	CH(CH_3_)_2_	82.93 ± 2.62	0.46 ± 0.07	38.57 ± 3.69
**3e**	CH_3_	CH_3_	53.17 ± 5.22	4.40 ± 0.15	36.61 ± 1.50
**3f**	H	C_6_H_5_	64.29 ± 5.07	1.10 ± 0.08	36.55 ± 3.07
**3g**	H	C_6_H_4_*p*-Br	86.96 ± 2.17	0.18 ± 0.03	38.60 ± 1.39
**3h**	-(CH_2_)_5_-		90.49 ± 1.31	0.07 ± 0.005	42.82 ± 0.96
**3i**	-(CH_2_)_3_-		71.25 ± 4.77	1.55 ± 0.10	46.33 ± 1.22
**Control**			90.42 ± 1.86 ^a^	0.08 ± 0.006 ^a^	55.22 ± 0.13 ^a^ 46.82 ± 3.75 ^b^

^a^ for carbenoxolone, ^b^ for 11β-glycyrrhetinic acid.

## Data Availability

Data available from the authors.

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
