# Peer review of "Synthesis of Novel 2-(Cyclopentylamino)thiazol-4(5H)-one Derivatives with Potential Anticancer, Antioxidant, and 11β-HSD Inhibitory Activities"

_ijms, 2023, doi:10.3390/ijms24087252_

Round 1

Reviewer 1 Report

The present original research article evaluates new compounds with anticancer, antioxidant and those that reveal by inhibition of 11β-hydroxysteroid dehydrogenase type 1. The topic is relevant, but there is a need for major changes for the improvement of the initial form:

Shape suggestions

The title should be revised because it is not appropriate in terms of repeating some words in it.

An extensive revision of the entire manuscript is needed at the level of grammar, typesetting, and punctuation.

Abbreviations are explained when they first appear in the abstract or main text and contribute to making the text easier to read and the information conveyed more efficiently. Once an abbreviation has been established and explained, it will be used throughout the entire manuscript, with the exception of the abstract, where it must be treated separately. Please revise the whole manuscript and explain the abbreviations used directly, without the explanation (e.g., L40 NAD, NADP etc.).

L53-74; L110-135- the information is organized in the form of an overly long paragraph, which decreases readability and comprehension. Please reorganize into shorter paragraphs that will be more logical and easier to understand.

L534- please remove “In conclusion, …”, as it is obvious. It is about Conclusions section.

The title of Table 1 should be placed after the synthesis scheme which should form a separate figure.

Content suggestions

The last sentence/sentence of the abstract should contain concluding/future direction information.

The aim of the article should be clarified separately in the last paragraph of the introduction section. The authors have described what they have done in the study, but they have not presented the novelty/special aspects it brings to the field, or the reason for choosing this topic. What was done in the research is already in the manuscript and is also provided in other sections.

The entities for which interactions are simulated in the molecular docking study should be better detailed and explained (PDB code for proteins, implications of hydrophobicity etc.).

In the discussion section, issues related to the results obtained in comparison with other studies that have been conducted on this topic should be included and detailed. In addition, it is important that in the final paragraphs of the Discussion section, the strengths of the present study and especially the limitations (and how these could be addressed in future research directions) be clearly stated and detailed.

Author Response

We would like to thank the Reviewer for the valuable remarks and comments, which significantly helped us to improve our manuscript. Below we present point-by-point responses to the comments.

The present original research article evaluates new compounds with anticancer, antioxidant and those that reveal by inhibition of 11β-hydroxysteroid dehydrogenase type 1. The topic is relevant, but there is a need for major changes for the improvement of the initial form:

Shape suggestions

1) The title should be revised because it is not appropriate in terms of repeating some words in it.

The title hasbeen changed in the manuscript.

2) An extensive revision of the entire manuscript is needed at the level of grammar, typesetting, and punctuation.

Abbreviations are explained when they first appear in the abstract or main text and contribute to making the text easier to read and the information conveyed more efficiently. Once an abbreviation has been established and explained, it will be used throughout the entire manuscript, with the exception of the abstract, where it must be treated separately. Please revise the whole manuscript and explain the abbreviations used directly, without the explanation (e.g., L40 NAD, NADP etc.).

All abbreviations used in the manuscript are explained.

3) L53-74; L110-135- the information is organized in the form of an overly long paragraph, which decreases readability and comprehension. Please reorganize into shorter paragraphs that will be more logical and easier to understand.

It has been corrected in the manuscript

4) L534- please remove “In conclusion, …”, as it is obvious. It is about Conclusions section.

It has been corrected in the manuscript.

5) The title of Table 1 should be placed after the synthesis scheme which should form a separate figure.

It has been corrected in the manuscript.

Content suggestions:

6) The last sentence/sentence of the abstract should contain concluding/future direction information.

The abstract has been rewritten.

7) The aim of the article should be clarified separately in the last paragraph of the introduction section. The authors have described what they have done in the study, but they have not presented the novelty/special aspects it brings to the field, or the reason for choosing this topic. What was done in the research is already in the manuscript and is also provided in other sections.

It has been changed in the manuscript.

8) The entities for which interactions are simulated in the molecular docking study should be better detailed and explained (PDB code for proteins, implications of hydrophobicity etc.).

The reason for using five different protein structures and brief discussion of the associated scatter of binding energies is introduced at the beginning of the Molecular docking section (2.2.2). The PDB codes of all structures relevant to our study (used for the current study and for procedure validation carried out previously) are given in the Methods section along with the link to the PDB database. We also added the XRD resolutions of the structures to show their applicability as molecular targets in ligand-protein docking (also Methods section). The issue of ligand hydrophobicity was discussed in more details in the Molecular docking section (2.2.2) to conclude that this parameter (as log P) is an important component to ligand-protein affinity, but more detailed, docking-based study still provided much more satisfactory agreement of the theoretical parameters (here: binding energies) with the experimental IC50 values.

9) In the discussion section, issues related to the results obtained in comparison with other studies that have been conducted on this topic should be included and detailed. In addition, it is important that in the final paragraphs of the Discussion section, the strengths of the present study and especially the limitations (and how these could be addressed in future research directions) be clearly stated and detailed.

The obtained results were compared with those obtained earlier.

Reviewer 2 Report

The aim of this study was to synthesise thiazolone derivatives with potential anticancer activity. Some existing 11β-HSD1 inhibitors were taken as a model, their inhibitory activity was described first,  and one structure 3h was found to have significant sub uM activity.   The in vitro activity against tumour cell lines was minimal, with no significant activity below 100 uM. The results from the ROS assays are difficult to interpret because they were done after 3-day exposure to the compounds at an extremely high dose 500 uM, conditions under which the cell numbers might have been induced or depleted significantly.

The authors should focus on 3h as a lead 11β-HSD1 inhibitor. Much of the introduction about anticancer mechanisms could be omitted. Fig. 3 could be replaced with a table of IC50. Fig 4 should be omitted, or at best focus on several cell lines with the most marked changes and repeated at lower, more realistic doses and much shorter treatment times.

3.7 What was the cell density?

3.8 How long was the treatment with compounds?

Author Response

We would like to thank the Reviewer for the valuable remarks and comments, which significantly helped us to improve our manuscript. Below we present point-by-point responses to the comments.

The aim of this study was to synthesise thiazolone derivatives with potential anticancer activity. Some existing 11β-HSD1 inhibitors were taken as a model, their inhibitory activity was described first,  and one structure 3h was found to have significant sub uM activity.   The in vitro activity against tumour cell lines was minimal, with no significant activity below 100 uM. The results from the ROS assays are difficult to interpret because they were done after 3-day exposure to the compounds at an extremely high dose 500 uM, conditions under which the cell numbers might have been induced or depleted significantly.

We much appreciate the Reviewer’s comment. ROS/RNS/GSH assays were performed at 500 µM to observe toxicity and potential anticancer effects. In fact, the fluorescence signal values were not related to the number of cells. Moreover, our research group successfully applied the selected methodology in several published papers (Sołek P. et al. Free Radical Biology and Medicine, 2022, 180, 153-164, https://doi.org/10.1016/j.freeradbiomed.2022.01.011, Solek P. et al. Apoptosis, 2019, 24(9-10), 773-784, doi: 10.1007/s10495-019-01557-5). Others also use high-concentration ranges of anticancer compounds in vitro (Kisacam MA. Naunyn Schmiedebergs Arch Pharmacol., 2023, 396(3), 547-555, doi: 10.1007/s00210-022-02354-9, Kullenberg F. et al. Cells. 2021 10(7),1717, doi: 10.3390/cells10071717.). The authors wish to emphasize that we are aware of the limitations of in vitro research. At the same time, we present here only preliminary results on the potential toxicity of selected 11β-HSD1 inhibitors. We are conducting further mechanistic studies to fully investigate the effects of the 11β-HSD1 inhibitors on cancer and normal cells, but these results are well beyond the scope of this manuscript. We also agree that our in vitro activity against tumor cell lines was minimal, with no significant effect below 100 µM. For this reason, our future research will include optimization in the research for the most optimal candidate structure.

The authors should focus on 3h as a lead 11β-HSD1 inhibitor. Much of the introduction about anticancer mechanisms could be omitted.

It has been corrected in the manuscript

Fig. 3 could be replaced with a table of IC50.

We much appreciate the Reviewer's comment. Figure 3 reflects the screening of the metabolic activity of various tumor and normal cells over a wide range of doses of the compounds tested. The authors presented the results as graphs because it will be more convenient for readers to understand the difference between the groups. In addition, this form of results presentation potentially encourages new readers to delve deeper into the manuscript. For this reason, the authors would like to stay with these graphs, because they attractively represent the basic anticancer activity of compounds tested.

Fig 4 should be omitted, or at best focus on several cell lines with the most marked changes and repeated at lower, more realistic doses and much shorter treatment times.

We much appreciate the Reviewer's comment. Our goal was to screen the compounds as broadly as possible, hence a large number of cell lines with different histology. In addition, based on the data presented here, we selected lines that were potentially most sensitive to the tested compounds. At the same time, the authors would like to emphasize that we presented here only preliminary results, therefore, in the future, the mechanism of 11β-HSD1 inhibitor and other new substances will be more developed. The authors are also familiar with the fact that in vitro systems are only simplified models and often the results are not clinically translated, therefore, the formulation of far-reaching conclusions may require more detailed research. This topic is extremely interesting for our team, therefore we believe the Reviewer's comment is very accurate and our future research will include new essays on realistic doses and shorter treatment times.

3.7 What was the cell density?

We appreciate the Reviewer's comment. Cell density was reported in the Materials and Methods section: 3.6 Cell culture

“All cell lines were cultured in T75 cell culture flasks or cell culture dishes to reach 80–90% confluency and seeded at a standard density of 5 × 103 cells/0.32cm2, except for BJ cells at 2 × 103 cells/0.32cm2.”

3.8 How long was the treatment with compounds?

We appreciate the Reviewer's comment. Exposure times with compounds were reported in the Materials and Methods section: 3.7 Metabolic activity

“The various concentrations of compounds were prepared and treated for the next 72 h”.

 In the case of the 3.8 Antioxidant activity section, the incubation time has been added to the description:

“After compounds treatment (72h), the cells were probed at a final concentration of 5 µM each for 15 minutes at 37°C.

Round 2

Reviewer 1 Report

The authors have significantly improved the manuscript based on the suggestions received.

Author Response

We would like to thank the Reviewer for the valuable remarks and comments, which significantly helped us to improve our manuscript.

Reviewer 2 Report

The data in Fig. 4 would be more compact and readable as a table.

Author Response

Comment: The data in Fig. 4 would be more compact and readable as a table.

Answer: We much appreciate the comment and we understand the Reviewer's point of view. Our group agrees with the view that “Graphs and Charts are more attractive and easy to understand than tables enable the reader to ‘see’ patterns in the data are easy to use for comparisons”. The data presented in Fig 4 illustrate the screening of multiple compounds on various cell lines. However, we added a line to indicate the level of control to make the results clearer. As suggested, we've also added a table reflecting the numerical results in the Supplementary Information (SI) section.